# Exosomal MicroRNA Analyses in Esophageal Squamous Cell Carcinoma Cell Lines

**DOI:** 10.3390/jcm11154426

**Published:** 2022-07-29

**Authors:** Sora Kim, Gwang Ha Kim, Su Jin Park, Chae Hwa Kwon, Hoseok I, Moon Won Lee, Bong Eun Lee

**Affiliations:** 1Department of Convergence Medical Sciences, Graduate School of Medicine, Pusan National University, Yangsan 50612, Korea; ssora0919@gmail.com; 2Department of Internal Medicine, Pusan National University College of Medicine, Busan 49241, Korea; neofaceoff@hanmail.net (M.W.L.); bonsul@hanmail.net (B.E.L.); 3Biomedical Research Institute, Pusan National University Hospital, Busan 49241, Korea; soo-jean@hanmail.net (S.J.P.); chkwon@pusan.ac.kr (C.H.K.); 4Department of Thoracic Surgery, Pusan National University College of Medicine, Busan 49241, Korea; ihoseok@pusan.ac.kr

**Keywords:** esophageal squamous cell carcinoma, exosome, microRNA, biomarker

## Abstract

Exosomal miRNAs have been studied in various cancers as minimally invasive biomarkers. This study aimed to investigate the potential of exosomal microRNAs (miRNAs) as biomarkers for esophageal squamous cell carcinoma (ESCC). Exosomes were isolated from cultures of esophageal epithelial cell and ESCC cell lines using ExoDisc, and exosomal miRNAs were detected via miRNA sequencing. Of the differentially expressed 14 miRNAs, the top 2 up-regulated miRNAs (miR-205-5p and miR-429) and top 2 down-regulated miRNAs (miR-375-3p and miR-483-3p) were selected as ESCC target miRNAs. Four selected exosomal miRNAs were validated in the plasma of 20 healthy controls (HCs) and 40 ESCC patients via quantitative reverse transcription-polymerase chain reaction. The expression of plasma exosomal miR-205-5p and miR-429 significantly increased, while that of plasma exosomal miR-375-3p was significantly reduced in ESCC patients compared to that in HCs. At cut-off values of 5.04, 2.564, and 0.136, the sensitivity and specificity for the diagnosis of ESCC were 72.5% and 70.0% for miR-205-5p, 60.0% and 60.0% for miR-429, and 65.0% and 65.0% for miR-375-3p, respectively. Based on the exosomal miRNAs identified in ESCC cell lines, our study demonstrated that plasma exosomal miR-205-5p, miR-429, and miR-375-3p could serve as potential biomarkers for ESCC diagnosis.

## 1. Introduction

Esophageal cancer (EC) is the 10th most common cancer worldwide and the sixth leading cause of cancer-related deaths [1]. The 5-year survival rate of patients with EC is reported to be 15–25% [2]. Histopathologic types of EC are mainly classified as adenocarcinoma and squamous cell carcinoma (SCC); these two types have a similar clinical manifestation but different features, including tumor location and risk factors [3]. Esophageal SCC (ESCC) is the most common EC in Asia, and it accounts for more than 90% of EC cases in Korea [4]. Patients with ESCC have a high mortality rate due to the advanced stage at diagnosis [5]. Therefore, the early detection and diagnosis of ESCC are critical for its treatment and prognosis [6].

Currently, endoscopy with forceps biopsy is the standard diagnostic modality for ESCC. However, it has several drawbacks, such as invasiveness, high cost, sampling error, and the possibility of missing small lesions. Although non-invasive blood tumor markers, such as carcinoembryonic antigen and SCC antigen, have been used for the detection and prognosis of ESCC in clinical settings, they are inadequate in the early diagnosis and assessment of tumor progression [7,8].

Liquid biopsy is a method of analyzing cancer-derived substances in various body fluids such as the blood, saliva, and cerebrospinal fluids [9]. A variety of analytes, including exosomes, microRNAs (miRNAs), circulating tumor cells, circulating cell-free DNAs, proteins, and various metabolites, are found in liquid biopsy specimens [10]. Of them, exosomes are membranous vesicles with a diameter of 30–200 nm and comprise nucleic acids, proteins, and lipids [11]. Exosomes are released from several cell types, such as blood cells, endothelial cells, and immune cells, and can regulate the biological activities of recipient cells by transporting miRNAs, lipids, proteins, and nucleic acids while circulating in the extracellular space [12]. These properties of exosomes can make them useful as tumor biomarkers.

miRNAs are a type of small noncoding RNAs, 17–24 nucleotides in length [13], that are involved in various biological activities, including cell proliferation, differentiation, and migration as well as disease initiation and progression [12]. Notably, miRNAs are aberrantly expressed in tumors and act as oncogenes or tumor suppressors, and it has been suggested that they can act as tumor biomarkers associated with tumorigenesis and development [14,15,16]. Recently, miRNAs have been identified in exosomes [12]. They are taken up by neighboring or distant cells and act as intercellular communication mediators [17]. The amount and composition of exosomal miRNAs are different between healthy controls (HCs) and diseased patients [18]. In addition, exosomal miRNAs are stable, as they are protected from endogenous ribonuclease activity [19]. Accordingly, the expression of exosomal miRNAs secreted by the tumor cells can be used as a less invasive biomarker that is indicative of the status of the tumor [20]. Therefore, we aimed to investigate exosomal miRNAs derived from human esophageal epithelial cell and human ESCC cell lines and validate the identified exosomal miRNAs in the plasma of HCs and ESCC patients.

## 2. Methods

### 2.1. Sample Preparation

#### 2.1.1. Cell Culture

A human esophageal epithelial cell line (Het-1A; ATCC, Rockville, MD, USA) and two human ESCC cell lines (TE8 and TE9; Riken Cell Bank, Tsukuba, Japan) were used in the present study. Het-1A cells were cultured in a BEGM medium (Lonza, Basel, Switzerland) supplemented with 10% fetal bovine serum (FBS) (Gibco, Grand Island, NY, USA) and 1% penicillin (Sigma, Burlington, MA, USA). TE8 and TE9 cells were cultured in an RPMI1640 medium (Gibco, Grand Island, NY, USA) supplemented with 10% FBS and 1% penicillin. The cells were maintained in an incubator at 37 °C with 5% CO_2_.

#### 2.1.2. Patient Blood Sampling

Peripheral blood samples were collected from 20 HCs and 40 ESCC patients at the Pusan National University Hospital (Busan, Korea) between May 2016 and April 2021. Peripheral blood samples (3 mL) were centrifuged at 800× *g* and 15 °C for 10 min to separate the plasma. Plasma samples were stored at −80 °C until exosome isolation.

The study protocol was approved by the Institutional Review Board of the Pusan National University Hospital (IRB No: H–1412–011–024), and written informed consent was obtained from all HCs and ESCC patients prior to sample collection. The study was performed in accordance with the Declaration of Helsinki.

### 2.2. Exosome Isolation

#### 2.2.1. ExoDisc Method for Cell Lines

When the cell density in 150 mm cell-culture dishes reached 90–100%, cells were incubated with a serum-free medium for 48 h. After incubation, 75 mL of cell culture supernatant medium was obtained and immediately centrifuged at 300× *g* for 10 min (cell removal). Then, centrifugation (2000× *g*, removal of dead cells) was performed for 10 min, followed by another centrifugation step carried out at 10,000× *g* for 30 min (removal of cell debris). The supernatant was concentrated with Amicon^®^ Ultra-15 (Millipore, MA, USA). Finally, exosomes were isolated via centrifugation using ExoDisc (Labspinner, Ulsan, Korea).

#### 2.2.2. ExoQuick Method for the Plasma

The plasma sample (300 μL) was transferred to a new clean tube and incubated for 5 min at 15 °C after the addition of 3 μL thrombin (System Biosciences, Mountain View, CA, USA). It was centrifuged at 10,000 rpm for 5 min; 250 μL of the supernatant was transferred to a new clean tube; 63 μL of ExoQuick solution (System Biosciences) was added. Next, it was incubated at 4 °C for 30 min, followed by centrifugation at 1500× *g* and at 4 °C for 30 min. The supernatant was then removed, and the sample mixture was centrifuged at 1500× *g* for 5 min to remove all liquids. The remaining exosome pellets were resuspended in 300 μL of 1× phosphate-buffered saline.

### 2.3. Characterization of Exosomes

#### 2.3.1. Transmission Electron Microscopy

The exosome sample was fixed at a ratio of 1:1 with 1% glutaraldehyde for 30 min. A 6 μL aliquot of this sample was pipetted onto a 100-mesh copper grid with a carbon-coated formvar film (EMS, Hatfield, PA, USA) and then incubated for 10 min. The excess liquid was removed by blotting. The grid was washed twice by brief contact with 100 μL distilled water, followed by blotting to remove excess liquid. Next, the grid was placed in 30 μL of 1% uranyl acetate for 12 s. After removing excess liquids by blotting, the sample was examined with a transmission electron microscope (JEM-1200EXII, JEOL, Akishima, Japan).

#### 2.3.2. Western Blotting

The exosomal protein concentration was measured using a BCA protein assay kit (Thermo Fisher Scientific, Waltham, WA, USA). Exosome samples containing the same amount of protein (50 μg) were separated via 4–15% sodium dodecyl sulfate polyacrylamide gel electrophoresis and transferred onto a polyvinylidene fluoride (PVDF) membrane. The PVDF membrane was then blocked with a 1% bovine serum albumin solution at 15 °C for 1.5 h and incubated overnight at 4 °C with one of the following primary antibodies: anti-CD63 (1:5000) (Abcam, Cambridge, UK), anti-TSG101 (1:5000) (Abcam), and HSP70 (1:5000) (Abcam). On the next day, the membrane was placed in Tris buffered saline-tween20 solution and washed at 15 °C for 1.5 h. The secondary antibody was then added to the membrane, and it was incubated at 15 °C. Finally, an enhanced chemiluminescence Western blotting kit was used to detect the exosome markers.

#### 2.3.3. Exo-Check Exosome Antibody Array

Lysis buffer (1×) was added to 50 μg of sample protein, followed by mixing at 15 °C for 30 min. Next, the sample mixture was filtered through a column, mixed with 5 mL of blocking buffer, and added to the membrane. Then, the membrane was incubated overnight at 4 °C on a rotary shaker. The next day, the membrane was washed twice, and 5 mL of detection buffer was added. The membrane was then incubated at 15 °C for 30 min on a rotary shaker. Finally, the membrane was washed thrice and developed.

### 2.4. RNA Sequencing

Total RNA isolated from each sample was used for constructing sequencing libraries with the SMARTer smRNA-Seq Kit for Illumina (Takara Bio, San Jose, CA, USA), according to the manufacturer’s protocol. In brief, input RNA was first polyadenylated to provide a priming sequence for an oligo-(dT) primer. cDNA synthesis was primed by the 3′-smRNA dT primer, which incorporated an adapter sequence at the 5′-end of each first-strand cDNA molecule. When the MMLV-derived PrimeScript™ Reverse Transcriptase reached the 5′-end of each RNA template, it added non-templated nucleotides that were bound by the SMRT smRNA Oligo-enhanced with locked nucleic acid technology for greater sensitivity. In the template-switching step, PrimeScript™ Reverse Transcriptase used the SMART smRNA Oligo as a template for the addition of a second adapter sequence to the 3′-end of each first-strand cDNA molecule. In the next step, full-length Illumina adapters (including index sequences for sample multiplexing) were added during the polymerase chain reaction (PCR) amplification. While the forward PCR primer was bound to the sequence added by the SMART smRNA Oligo, the reverse PCR primer was bound to the sequence added by the 3′-smRNA dT primer. The resultant cDNA molecule library included sequences required for clustering on an Illumina flow cell. The validation of the libraries was performed by checking their size, concentration, and purity on an Agilent Bioanalyzer (Agilent Technologies, Santa Clara, CA, USA). The libraries were pooled in equimolar amounts and sequenced on an Illumina HiSeq 2500 instrument to generate 51 base reads. Image decomposition and quality value calculations were performed by using the Illumina pipeline modules. 

### 2.5. Quantitative Reverse Transcription-Polymerase Chain Reaction

The expression of cellular exosomal miRNAs was assessed using quantitative reverse transcription-polymerase chain reaction (qRT-PCR). Total RNA was isolated from 400 μL exosome sample using the miVana PARISTM (Thermo Fisher Scientific, Waltham, WA, USA), according to the manufacturer’s instructions. cDNA synthesis was performed using the TaqMan miRNA RT kit (Applied Biosystems, Foster City, CA, USA). qRT-PCR was performed using TaqMan^®^ Universal PCR Master Mix II and monitored in real time using an RG-6000 real-time PCR. The cycle threshold (Ct) values were calculated using Rotor-Gene 6000 series software 1.7 (Corbett Research, Sydney, Australia). The relative expression levels of miRNAs in plasma samples were normalized with the 2^−ΔΔCT^ method using miR-191 as the internal control [21].

### 2.6. Bioinformatics Analysis

#### 2.6.1. Differential miRNA Expression

Cellular exosomal miRNAs obtained in a single cell culture experiment were analyzed for bioinformatics. Raw data (the reads for each miRNA) were normalized using the trimmed mean of the M-value method with edgeR. For pre-processing, miRNAs with a zero-count in more than one sample were excluded, leaving 350 mature miRNAs for downstream analysis. A dummy variable “1” was added to the normalized read count of the filtered miRNAs to facilitate log2 transformation for plotting. For each miRNA, log counts per million and log fold change were calculated between human esophageal epithelial cell and ESCC cell lines. A statistical hypothesis test for a comparison of the two groups was conducted using the exact test in edgeR. Differentially expressed miRNAs between the two groups were chosen using the following criteria: |fold change (FC)| ≥ 2 and *p* < 0.05. Volcano plots of the differentially expressed miRNAs were generated using edgeR.

#### 2.6.2. Hierarchical Clustering

Hierarchical clustering analyses were performed using complete linkage and Euclidean distances as a measure of similarity to display the expression patterns of differentially expressed miRNAs that satisfied the criteria |FC| ≥ 2 and *p* < 0.05.

### 2.7. Statistical Analysis

Data are expressed as median and interquartile range (IQR). The Mann–Whitney U test was used to compare exosomal miRNA expression levels between HCs and ESCC patients. Receiver-operating characteristic (ROC) curves and the areas under the curve (AUCs) were used to further evaluate the levels of the four selected miRNAs to differentiate ESCC patients from HCs. The sensitivity, specificity, and positive and negative predictive values of the miRNAs for differentiating ESCC patients from HCs were expressed using 95% confidence intervals (CIs). All statistical analyses were performed using IBM Statistical Package for the Social Sciences (SPSS) version 23.0 for Windows (IBM Co, Armonk, NY, USA). All *p*-values were two-sided, and statistical significance was set at *p* < 0.05.

## 3. Results

### 3.1. Isolation and Characterization of Exosomes

The first step was to capture and identify exosomes derived from Het-1A, TE8, and TE9 cell lines. Exosomes were isolated from conditioned media of Het-1A, TE8, and TE9 using ExoDisc (Figure 1). The diameter of the exosomes was approximately 30–200 nm, with cup-shaped morphology, indicating the typical characteristics of exosomes (Figure 2). Next, the expression of exosome surface markers was assessed via Western blotting; the expression of exosome markers CD63, TSG101, and HSP70 was detected in all cell lines (Figure 3A). Additionally, an Exo-Check exosome antibody array was conducted to assess other exosome markers (FLOT1, ICAM, ALIX, CD81, CD63, EpCAM, ANX5, and TSG101); the positive control marker was also detected (Figure 3B).

### 3.2. Analysis of Diffrentially Expressed Exosomal miRNAs in Esophageal Squamous Cell Carcinoma Cell Lines

miRNA sequencing was performed to confirm the expression patterns of exosomal miRNAs in ESCC cell lines. A total of 2656 miRNAs were identified; bioinformatic analysis was performed based on 423 miRNAs as 2233 miRNAs had a 0-count value. When the 423 identified miRNAs were filtered by |FC| ≥ 2 and *p* < 0.05, 14 differentially expressed miRNAs (five up-regulated and nine down-regulated) were detected in the ESCC cell lines compared to those in the esophageal epithelial cell line (Figure 4A, Table 1). To confirm the expression pattern of these 14 miRNAs, a hierarchical clustering analysis was performed, based on which the exosomal miRNA expression patterns of esophageal epithelial cell and ESCC cell lines were divided into two groups (Figure 4B)

### 3.3. Verification of Target Exosomal miRNAs in Patients with Esophageal Squamous Cell Carcinoma

Among the 14 differentially expressed miRNAs, the top two up-regulated miRNAs (miR-205-5p and miR-429) and top two down-regulated miRNAs (miR-375-3p and miR-483-3p) were selected as ESCC target miRNAs. Their expression was verified in plasma exosomal miRNAs obtained from 20 HCs and 40 ESCC patients.

The expression levels of plasma exosomal miR-205-5p and miR-429 significantly increased in ESCC patients compared to those in HCs (57031.725 [IQR 3.071, 179.540] vs. 780.200 [IQR 0.049, 5.926], *p* = 0.001 and 413.874 [IQR 1.190, 17.005] vs. 2.355 [IQR 0.263, 3.805], *p* = 0.012, respectively) (Figure 5A,B). In contrast, the expression of plasma exosomal miR-375-3p was significantly reduced in ESCC patients compared to that in HCs (0.201 [IQR 0.038, 0.174] vs. 11.838 [IQR 0.064, 15.414], *p* = 0.002) (Figure 5C). However, there was no difference in the expression of plasma exosomal miR-483-3p between ESCC patients and HCs (1.955 [IQR 0.710, 2.746] vs. 1.544 [IQR 0.554, 2.147], *p* = 0.335) (Figure 5D). 

The ROC curves were analyzed to determine whether plasma exosomal miRNAs could be used as potential diagnostic biomarkers to differentiate ESCC patients from HCs (Figure 6). The AUCs were 0.770 (95% CI 0.633–0.907, *p* = 0.001) for miR-205-5p, 0.699 (95% CI 0.569–0.829, *p* = 0.012) for miR-429, 0.741 (95% CI 0.591–0.591, *p* = 0.002) for miR-375-3p, and 0.577 (95% CI 0.422–0.731, *p* = 0.335) for miR-483-3p. At cut-off values of 5.04, 2.564, 0.136, and 1.513, the sensitivity and specificity for the diagnosis of ESCC were 72.5% and 70.0% for miR-205-5p, 60.0% and 60.0% for miR-429, 65.0% and 65.0% for miR-375-3p, and 55.0% and 55.0% for miR-483-3p, respectively.

## 4. Discussion

The reported expression profiles of various blood miRNAs in ESCC are inconsistent. In the present study, exosomal miRNAs were extracted from esophageal epithelial cell and ESCC cell lines, and two up-regulated miRNAs (miR-205-5p and miR-429) and two down-regulated miRNAs (miR-375-3p and miR-483-3p) were selected based on the difference in expression in both the cell lines. We found that plasma exosomal miR-205-5p, miR-429, and miR-375-3p could act as potential biomarkers for the diagnosis of ESCC. In particular, miR-205-5p and miR-375-3p showed high sensitivity and specificity for differentiating ESCC patients from HCs. To our knowledge, this is the first study to investigate exosomal miRNA expression profiles in esophageal epithelial cell and ESCC cell lines.

Cellular gene products, such as miRNAs, mRNAs, and proteins, are reported to be packaged inside exosomes and delivered to recipient cells to exert their biological effects [22]. Among them, exosomal miRNAs can act as biomarkers owing to their advantages in terms of quantity, quality, and stability over non-exosomal miRNAs [23]. In addition, exosomal miRNAs are regarded as potential biomarkers for human cancer in many studies because of their features such as stability, tissue-specific expression, and secretion into all biological fluids [19]. However, exosomes in the blood are known to be derived from various cells. Their heterogeneous origins might limit the detection of tumor-specific exosomes in peripheral blood samples. In addition, the vast number of exosomes from different cell types can dilute the exosome population of tumor cells and significantly reduce the proportion of tumor-derived miRNAs in the sequencing library [24]. In contrast, since cell line-derived exosomes are derived from one cell type, they will have miRNAs that are specifically expressed in that cell. In addition, a large number of exosomes can be acquired via cell culture in the conditioned media compared to the number of exosomes in the patient’s blood sample. In two recent studies that investigated exosomes in cell lines, colorectal cancer cell-derived exosomes could improve the migration of normal colonic epithelial cells [25] and exosomes from metastatic breast cancer cell lines were found to be rich in related proteins capable of inducing cancer metastasis [26].

In the present study, of the 14 differentially expressed exosomal miRNAs found in esophageal epithelial cell and ESCC cell lines, 2 up-regulated miRNAs (miR-205-5p and miR-429) and 2 down-regulated miRNAs (miR-375-3p and miR-483-3p) were selected to evaluate their clinical application in the diagnosis of ESCC. The expression of plasma exosomal miR-205-5p and miR-429 significantly increased and that of plasma exosomal miR-375-3p was significantly reduced in ESCC patients compared to that in HCs. However, the expression of plasma exosomal miR-483-3p did not differ between ESCC patients and HCs.

miR-205-5p has been reported to promote tumor initiation, progression, and resistance to anti-tumor therapy in various cancers [27,28]. miR-205-5p reportedly shows increased levels in the cancer tissues and sera of patients with non-small cell lung cancer [29], and it suppresses the invasiveness of cancer cells in oral SCC [30]. In the present study, exosomal miR-205-5p was significantly up-regulated in the plasma of ESCC patients as well as ESCC cell lines, which is consistent with the results of a recent study showing that the expression of circulating miR-205-5p is higher in ESCC patients than in HCs [31].

miR-429 is aberrantly expressed in several cancers and has diverse roles; it acts as a tumor suppressor in gastric cancer [32] but exerts an oncogenic effect in colorectal cancer [33]. In hepatocellular carcinoma, conflicting results pertaining to miRNAs and exosomal miRNAs have been reported [34,35]. In the present study, exosomal miR-429 was significantly up-regulated in both ESCC cell lines and the plasma of ESCC patients. These results are contrary to the results of previous studies stating that miR-429 might act as a tumor suppressor in ESCC [36,37]. However, in those studies, exosomal miR-429 was not measured as in our study. These conflicting results may be due to the differences in sample types, miRNA detection methods, and a lack of a common miRNA internal control [34]. Furthermore, considering that the interpretation of miRNA-related data is very complex, the molecules and signals involved in the regulation of miRNAs, the precise roles of circulating miRNAs in oncogenesis, and the large heterogeneity of miRNAs in each cancer type and at different tumor stages require further investigation [38].

miR-375-3p has been reported to be commonly down-regulated in several types of cancer, including gastric cancer and hepatocellular carcinoma [39]. Reportedly, miR-375 acts as a tumor suppressor through the inhibition of cell proliferation, colony-forming ability, and the metastasis of ESCC in vitro and in vivo [40], and its expression is reduced in the plasma of ESCC patients compared to that in the plasma of HCs [41]. Similarly, in the present study, exosomal miR-375-3p was significantly down-regulated in the plasma of ESCC patients as well as ESCC cell lines.

miR-483-3p is known to be expressed in different forms in various types of cancers [42]. In pancreatic cancer, miR-483-3p is overexpressed, which is related to the short survival rate [43]. In contrast, miR-483-3p is down-regulated in hepatitis B virus-associated hepatocellular carcinoma [44]. In the present study, there was no difference in the expression of plasma exosomal miR-483-3p between ESCC patients and HCs.

This study has some limitations that should be addressed in future studies. First, exosomal miRNAs identified in ESCC cell lines were not validated in cancer tissues. Second, since we focused on the role of exosomal miRNAs as biomarkers for differentiating ESCC patients from HCs, we did not analyze the clinicopathological features of ESCC patients and long-term follow-up data, such as chemotherapeutic response and survival, in the present study. Third, the number of ESCC patients included in the present study was relatively small, as this was a single-institution-based study. Finally, exosomal miRNAs detected in the plasma of ESCC patients are not organ specific. Although we tried to exclude other current malignancies via a detailed work-up, there is still a possibility that undetected malignancies are present in other organs, which could be a source of exosomal miRNAs. Therefore, a multi-center study with a larger number of ESCC patients is needed to validate the role of plasma exosomal miRNAs as biomarkers for ESCC, as shown in the present study.

In conclusion, we observed a higher expression of plasma exosomal miR-205-5p and miR-429 and a lower expression of plasma exosomal miR-375-3p in ESCC patients than in HCs, based on the exosomal miRNAs identified in esophageal epithelial cell and ESCC cell lines. Our results suggest that plasma exosomal miRNAs could act as potential non-invasive biomarkers for the detection of ESCC. The role of these miRNAs should be further explored in large, prospective, multi-center long-term follow-up studies.

## Figures and Tables

**Figure 1 jcm-11-04426-f001:**
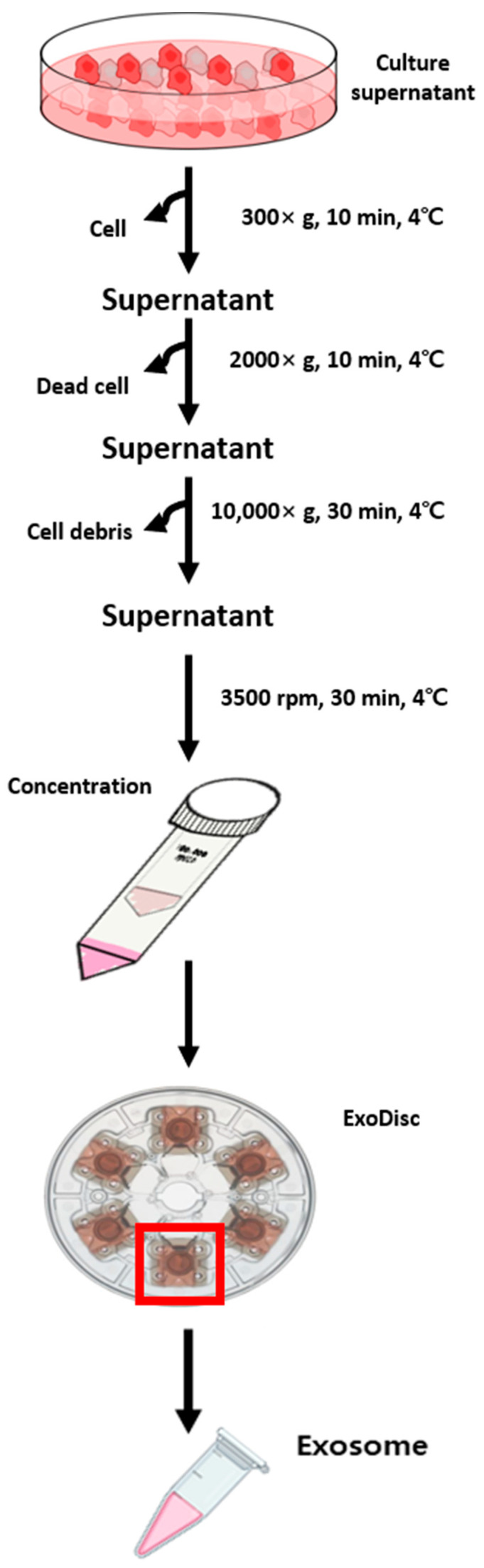
Exosome isolation from the cell culture supernatant using ExoDisc.

**Figure 2 jcm-11-04426-f002:**
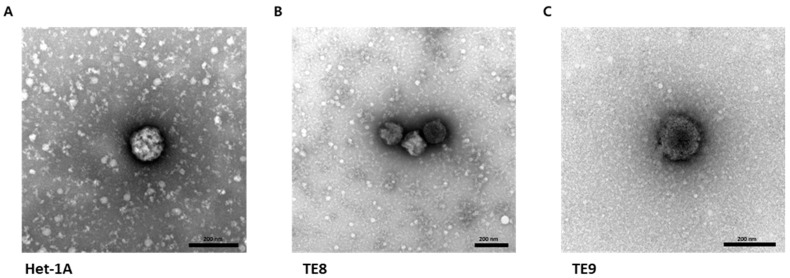
Images of esophageal cell line-derived exosomes. Representative transmission electron microscopic images of cellular exosomes isolated from Het-1A (**A**), TE8 (**B**), and TE9 (**C**) cell lines. Scale bar = 200 nm.

**Figure 3 jcm-11-04426-f003:**
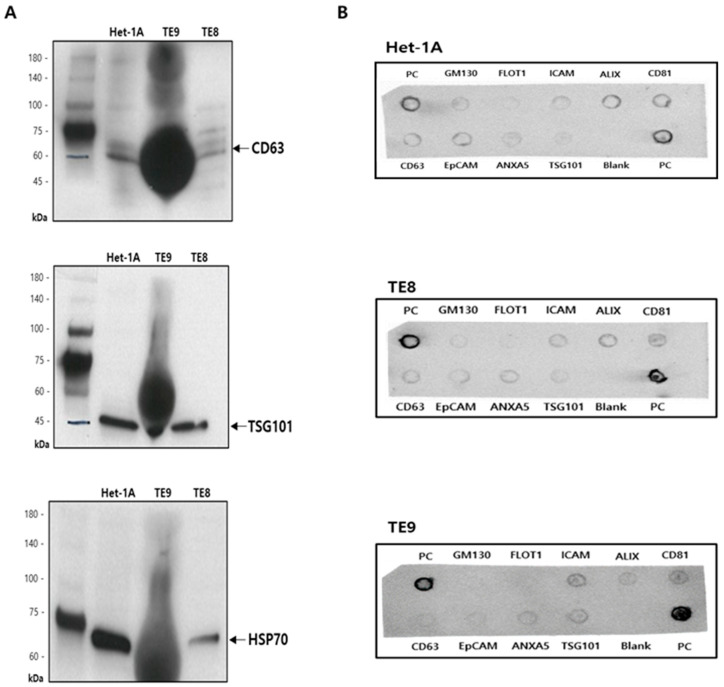
Characterization of exosomes in esophageal epithelial cell and esophageal squamous cell carcinoma cell lines. (**A**) Expression of exosome markers detected with Western blot analysis (CD63, TSG101, and HSP70). (**B**) Representative Exo-Check exosome antibody array for detecting exosomal markers (FLOT1, ICAM, ALIX, CD81, CD63, EpCAM, ANXA5, and TSG101) and for assessing cellular contamination (Golgi marker GM130) and positive control (PC).

**Figure 4 jcm-11-04426-f004:**
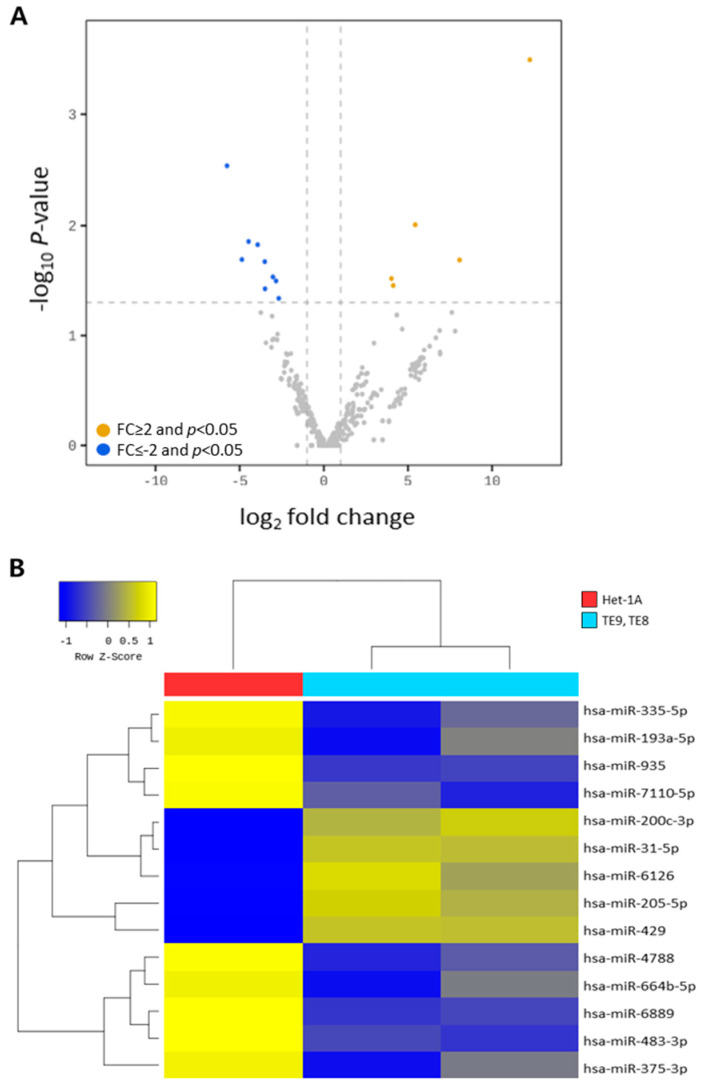
Differentially expressed exosomal miRNAs between esophageal epithelial cell and esophageal squamous cell carcinoma cell lines. (**A**) Volcano plots of differentially expressed exosomal miRNAs. Colored circles represent miRNAs significant at |fold change (FC)| ≥ 2 and *p* < 0.05. Orange dot, up-regulated; blue dot, down-regulated. (**B**) Heatmap of hierarchical clustering based on 14 differentially expressed exosomal miRNAs (five up-regulated and nine down-regulated). Samples with similar miRNA expression were grouped.

**Figure 5 jcm-11-04426-f005:**
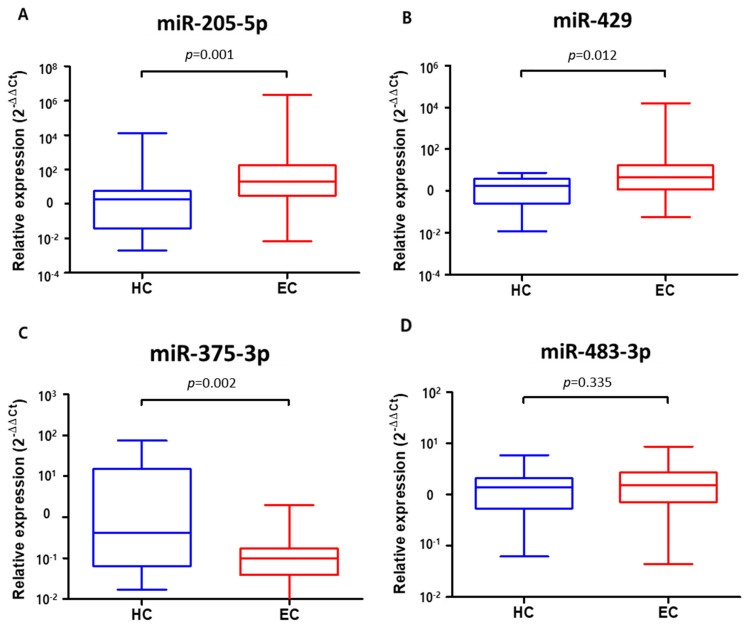
Expression levels of four plasma exosomal miRNAs in 20 healthy controls and 40 patients with esophageal squamous cell carcinoma. (**A**) miR-205-5p. (**B**) miR-429. (**C**) miR-375-3p. (**D**) miR-483-3p. Boxes represent interquartile ranges with bars representing minimum to maximum.

**Figure 6 jcm-11-04426-f006:**
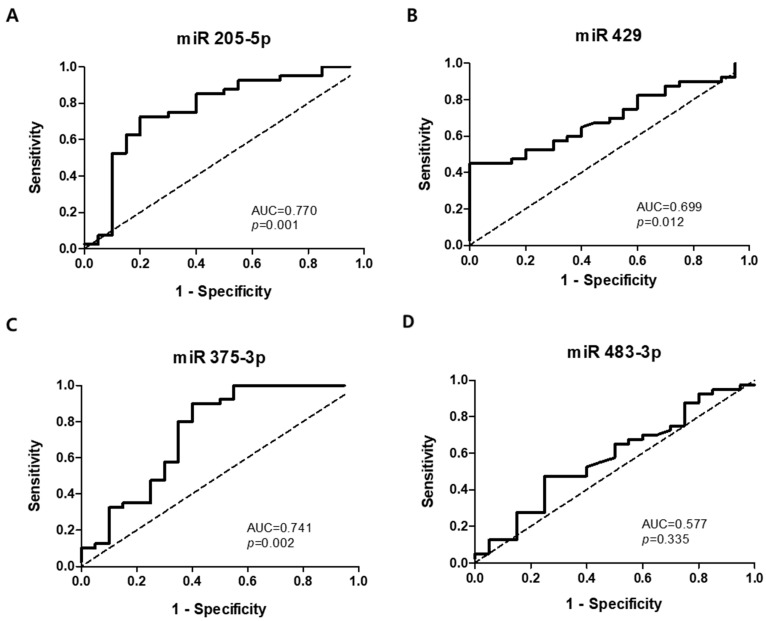
Receiver-operating characteristic curve analyses of four plasma exosomal miRNAs to differentiate patients with esophageal squamous cell carcinoma from healthy controls. (**A**) miR-205-5p. (**B**) miR-429. (**C**) miR-375-3p. (**D**) miR-483-3p. AUC, area under the curve.

**Table 1 jcm-11-04426-t001:** Five up-regulated miRNAs and nine down-regulated miRNAs in esophageal squamous cell carcinoma cell lines.

Mature miRNA	Family	Sequence	Fold Change	*p*-Value	Expression Level
hsa-miR-205-5p	hsa-mir-205	UCCUUCAUUCCACCGGAGUCUG	4843.079	<0.001	Up
hsa-miR-429	hsa-mir-429	UAAUACUGUCUGGUAAAACCGU	267.564	0.020	Up
hsa-miR-6126	hsa-mir-6126	GUGAAGGCCCGGCGGAGA	43.004	0.009	Up
hsa-miR-200c-3p	hsa-mir-200c	UAAUACUGCCGGGUAAUGAUGGA	17.440	0.035	Up
hsa-miR-31-5p	hsa-mir-31	AGGCAAGAUGCUGGCAUAGCU	16.186	0.030	Up
hsa-miR-193a-5p	hsa-mir-193a	UGGGUCUUUGCGGGCGAGAUGA	−6.395	0.045	Down
hsa-miR-335-5p	hsa-mir-335	UCAAGAGCAAUAACGAAAAAUGU	−7.156	0.031	Down
hsa-miR-935	hsa-mir-935	CCAGUUACCGCUUCCGCUACCGC	−8.137	0.029	Down
hsa-miR-664b-5p	hsa-mir-664b	UGGGCUAAGGGAGAUGAUUGGGUA	−11.285	0.037	Down
hsa-miR-7110-5p	hsa-mir-7110	UGGGGGUGUGGGGAGAGAGAG	−11.416	0.021	Down
hsa-miR-4788	hsa-mir-4788	UUACGGACCAGCUAAGGGAGGC	−15.234	0.014	Down
hsa-miR-6089	hsa-mir-6089-1 hsa-mir-6089-2	GGAGGCCGGGGUGGGGCGGGGCGG	−22.213	0.013	Down
hsa-miR-375-3p	hsa-mir-375	UUUGUUCGUUCGGCUCGCGUGA	−29.156	0.020	Down
hsa-miR-483-3p	hsa-mir-483	UCACUCCUCUCCUCCCGUCUU	−53.918	0.002	Down

## Data Availability

The data presented in this study are openly available in the BioProject database (http://www.ncbi.nlm.nih.gov/bioproject/853801: accessed on 1 July 2022).

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
