# Peer review of "Exosomal MicroRNA Analyses in Esophageal Squamous Cell Carcinoma Cell Lines"

_jcm, 2022, doi:10.3390/jcm11154426_

Round 1
Reviewer 1 Report
In the manuscript entitled " Exosomal MicroRNA Analysis in Esophageal Squamous Cell Carcinoma Cell Lines", Kim et al. demonstrated that plasma exosomal miR-205-5p, miR-429, and miR-375-3p could act as potential biomarkers for ESCC diagnosis.
The aim of the study is clear with an effort to demonstrate that the expression of plasma exosomal miR-205-5p and miR-429 was significantly up-regulated and that of plasma exosomal miR-375-3p was significantly down-regulated in ESCC patients compared to HCs. Analytics and results are based on esophageal epithelial cell and esophageal squamous cell carcinoma cell lines. Upregulation of miR-429 inhibits invasion and promotes apoptosis in esophageal carcinoma cells.
The manuscript is well written. I have numbers of questions and comments as follows.
Comments:
1. Sample size for miRNA seq data Is small. Figure 4: Clustering over 3 cell lines may not draw meaningful conclusion.
2. Data Availability Statement: miRNAseq data should be submitted to NCBI GEO
3. References #1 Need edit.
Author Response
Reply to Reviewer #1’s Comments
We would like to thank the reviewer for the constructive critique to improve the manuscript. We have made every effort to address the issues raised and respond to all comments. The revisions are indicated in red font in the revised manuscript. Please find below a detailed, point-by-point response to the reviewer's comments. We hope that our revisions meet the reviewer’s expectations.
- Sample size for miRNA seq data Is small. Figure 4: Clustering over 3 cell lines may not draw meaningful conclusion.
Response: As already stated in the Result section, a total of 2,656 miRNAs were identified; the bioinformatic analysis was performed based on 423 miRNAs, as 2,233 miRNAs had a 0-count value. When the 423 identified miRNAs were filtered by |FC| ≥ 2 and p < 0.05, 14 differentially expressed miRNAs (five up-regulated and nine down-regulated miRNAs) were detected. Therefore, we believe that our study would give meaningful results. To improve clarity and readability, we have revised the Figure 4 (B) legend as follows:
(B) Heatmap of hierarchical clustering based on 14 differentially expressed exosomal miRNAs (five up-regulated and nine down-regulated). Samples with similar miRNA expression were grouped.
- Data Availability Statement: miRNAseq data should be submitted to NCBI GEO.
Response: We have recently uploaded our raw data to the BioProject database and added data availability statement in the revised manuscript.
Data Availability Statement: The data presented in this study are openly available in the BioProject database (http://www.ncbi.nlm.nih.gov/bioproject/853801).
- References #1 Need edit.
Response: We have updated reference 1 as follows:
- Sung, H.; Ferlay, J.; Siegel, R.L.; Laversanne, M.; Soerjomataram, I.; Jemal, A.; Bray, F. Global cancer statistics 2020: GLOBOCAN estimates of incidence and mortality worldwide for 36 cancers in 185 countries. CA Cancer J Clin 2021, 71, 209-249, doi:10.3322/caac.21660.
Reviewer 2 Report
In this paper the authors aim to identify biomarkers for ESCC detection using an exosome based miRNA approach. Exosomes are thought to harbor several RNA components and have recently been rediscovered in the context of liquid biopsies. miRNA's are the principal fragments present in exosomes and the authors choose wisely to focus on these. To identify exosomes several steps need to be taken into account such as elektron microscopy and WB for known exosome markers, which the authors did successfully. Exosomes were derived from 2 ESCC and a normal esophageal cell line and miRNA profiles compared to identify altered miRNA's. In the second part of the study they compared these specific miRNA's in serum from ESCC patients and healthy volunteers.
Major comments
1. The authors do not verify if they have used biological triplicates of the cell-line experiments. This is essential to draw any conclusions on the detected miRNA's.
2. The differently altered miRNA's have not been validated in any tissue, e.g. they might not be ESCC specific in the patients. Are the authors able to provide this information? Can they go back to analyse the ESCC specimen?
3. The differences between the altered miRNA's are very subtle with large confidence intervals. This is also reflected in the low sensitivity and specificity of each miRNA for ESCC detection. How do the authors expect to be able to use this in clinical practice?
4. why did the authors use a cell-line discovery approach instead of a conventional discovery cohort in patients followed by a validation step in a second larger patient cohort?
Minor
1. The cell-lines have been grown to near confluence and afterwards 'starved' in serum-free medium for 2 days. The authors do not explain why they choose this experimental set-up. I wonder if they have tried different experimental conditions and if this set-up does not select for cells in growth arrest.
2. The exact mean miRNA levels for the ESCC patients can not be easily derived from the figures.
3. Although the authors mention in the limitations that they haven't analyzed the patient data, a table with at least basic information regarding the disease stage is required.
4. introduction, line: "although basic blood tumor markers...' lacks a reference
Author Response
Reply to Reviewer #2’s Comments
We would like to thank the reviewer for the constructive critique to improve the manuscript. We have made every effort to address the issues raised and respond to all comments. The revisions are indicated in red font in the revised manuscript. Please find below a detailed, point-by-point response to the reviewer's comments. We hope that our revisions meet the reviewer’s expectations.
- The authors do not verify if they have used biological triplicates of the cell-line experiments. This is essential to draw any conclusions on the detected miRNAs.
Response: Thank you for this suggestion. It would have been interesting to explore this aspect. Cell-line experiments are usually repeated thrice. However, owing to the high cost of analysis, it was inevitably not possible to use replicates for the experiment. Instead, we performed a validation study of cell-line miRNA results on blood samples of the patients. We believe that this process would eliminate the limitation of cell-line experiments in our study.
- The differently altered miRNA's have not been validated in any tissue, e.g. they might not be ESCC specific in the patients. Are the authors able to provide this information? Can they go back to analyze the ESCC specimen?
Response: We have already described the non-validation of our results in cancer tissues as one of limitations of this study, in the Discussion section. Additionally, we attempted to confirm our findings using tissue samples. Although we agree that this is an important consideration, it is beyond the scope of this manuscript because we could not obtain the paired tissue samples from the Korea Biobank network. We have recently started to collect the paired blood and tissue samples at our hospital. However, at present, it is impossible to validate our results in the tissue samples.
- The differences between the altered miRNA's are very subtle with large confidence intervals. This is also reflected in the low sensitivity and specificity of each miRNA for ESCC detection. How do the authors expect to be able to use this in clinical practice?
Response: We agree this is a potential limitation of the study due to the following reasons:
(1) a relatively small number of the subjects included in our study and (2) an inevitable limitation of serum biomarkers that the exosomal miRNAs detected in the plasma of ESCC patients are not organ specific. We have added this as a potential limitation in Discussion. The revised test reads as follows:
Finally, exosomal miRNAs detected in the plasma of ESCC patients are not organ specific. Although we tried to exclude other current malignancies via a detailed work-up, there is still a possibility that undetected malignancies are present in other organs, which could be a source of exosomal miRNAs.
- Why did the authors use a cell-line discovery approach instead of a conventional discovery cohort in patients followed by a validation step in a second larger patient cohort?
Response: First, there have been a few studies on miRNAs using ESCC cell lines compared to esophageal adenocarcinoma cell lines. Second, because the incidence of ESCC is low (10/100,000 persons) in Korea, it is difficult to collect blood samples from many ESCC patients. Another limitation was our limited research budget during the study period. Therefore, we sincerely request reviewer’s consideration.
- The cell-lines have been grown to near confluence and afterwards 'starved' in serum-free medium for 2 days. The authors do not explain why they choose this experimental set-up. I wonder if they have tried different experimental conditions and if this set-up does not select for cells in growth arrest.
Response: FBS may contain endogenous exosomes, which could result in false positivity of exosomes. Therefore, a serum-free medium is used to collect exosomes to minimize FBS contamination, such as various proteins and metabolites in the serum. Accordingly, in our study, cells were maintained in a serum-free medium, containing nutrients essential for cell growth for 24 to 48 h to produce exosomes.
- The exact mean miRNA levels for the ESCC patients cannot be easily derived from the figures.
Response: To improve the clarity of the figure, we fixed the vertical axis interval using logarithm. First, we planned to include the median value of each miRNA in the figure; however, it makes the figures more complicated. Therefore, we kept the data of the figures in their existing format.
- Although the authors mention in the limitations that they haven't analyzed the patient data, a table with at least basic information regarding the disease stage is required
Response: In the present study, we focused on the potential of plasma exosomal miRNAs to differentiate ESCC patients from healthy controls. Therefore, we omitted the specific details of ESCC patients. We are planning follow-up research on the additional roles of plasma exosomal miRNAs, such as treatment response to chemo/radiotherapy or prognostic markers, after including more ESCC patients.
- introduction, line: "although basic blood tumor markers...' lacks a reference
Response: We have added the following references in the Introduction.
Although non-invasive blood tumor markers, such as SCC antigen and carcinoembryonic antigen, have been used for the detection and prognosis of ESCC in clinical settings, they perform unsatisfactorily in early diagnosis and assessment of tumor progression [7, 8].
- Lukaszewicz-Zajac, M.; Mroczko, B.; Kozlowski, M.; Niklinski, J.; Laudanski, J.; Szmitkowski, M. Higher importance of interleukin 6 than classic tumor markers (carcinoembryonic antigen and squamous cell cancer antigen) in the diagnosis of esophageal cancer patients. Dis Esophagus 2012, 25, 242-249, doi:10.1111/j.1442-2050.2011.01242.x.
- Choi, M.K.; Kim, G.H.; I, H.; Park, S.J.; Lee, M.W.; Lee, B.E.; Park, D.Y.; Cho, Y.K. Circulating tumor cells detected using fluid-assisted separation technique in esophageal squamous cell carcinoma. J Gastroenterol Hepatol 2019, 34, 552-560, doi:10.1111/jgh.14543.
Round 2
Reviewer 2 Report
Many thanks for your detailed feedback and improvements to the paper. I would like a line added to the cell-line experiments in the methodology added which clarifies these have not been performed in triplicate.
Author Response
We sincerely thank the reviewer for kind comments. We have added the following sentence in the Method section (2.6.1. Differential miRNA Expression) according to the reviwer's comments as follows:
Cellular exosomal miRNAs obtained in a single cell culture experiment were analyzed for bioinformatics. Raw data (the reads for each miRNA) were normalized using the trimmed mean of the M-value method with edgeR.
